# Multi-fidelity Deep Symbolic Optimization

## Abstract

Although Symbolic Optimization (SO) can be used to model many challenging problems, the computational cost of evaluating large numbers of candidate solutions is intractable in many real-world domains for existing SO algorithms based on reinforcement learning (RL). While lower-fidelity surrogate models or simulations can be used to speed up solution evaluation, current methods for SO are unaware of the existence of the multiple fidelities and therefore do not natively account for the mismatch between lower and higher fidelities. We propose to explicitly reason over the multiple fidelities. For that, we introduce Multi-Fidelity Markov Decision Processes (MF-MDPs) and propose a whole new family of multi-fidelity SO algorithms that account for multiple fidelities and their associated costs. We conduct experimental evaluation in two challenging SO domains, Symbolic Regression and Antibody Optimization, and show that our methods outperform fidelity-agnostic and fidelity-aware baselines.

## 1 Introduction

Symbolic Optimization (SO) is the problem of searching over sequences of tokens to optimize a black-box reward function. Numerous challenging real-world problems can and have been solved using SO, including Symbolic Regression (Lu et al., 2016), Neural Architecture Search (Yu et al., 2020), and Program Synthesis (Kitzelmann, 2009). Deep Symbolic Optimization (DSO) (Petersen et al., 2021) models the token sampling process as a Reinforcement Learning (RL) problem. Although DSO has achieved remarkable success in varied domains, such as winning the real-world track in the SRBench symbolic regression competition (Kommenda et al., 2022), it makes the key assumption that the actual reward function is inexpensive enough that a large number of candidate solutions—typically hundreds of thousands—can be evaluated during training. However, this assumption does not hold in many real-world SO domains, such as Antibody Optimization (Norman et al., 2020) where even *in silico* simulations of antibody quality are computationally expensive (Barlow et al., 2018).

One way to cope with expensive evaluation is to rely on a computationally inexpensive but lower fidelity surrogate model of the actual reward. However, this incurs on a mismatch between the optimal solution for the original and the lower fidelity problems. This mismatch in the reward functions has typically been viewed as a Transfer Learning problem (Silva and Costa, 2019) in an ad hoc manner, where the learning process is carried out using the cheap simulation and then the resulting policy is either adapted (Hanna and Stone, 2017) or transferred directly (MacAlpine and Stone, 2018) to the desired domain. Other multi-fidelity methods focus on modifying the simulator, which is often not possible (Peherstorfer et al., 2018).

In this paper, we posit that it is more effective to explicitly reason over multiple reward functions in different fidelities, allowing more faithful modeling of the problem and more effective usage of a limited budget in the highest fidelity. We propose a new multi-fidelity framework that encapsulates many possible strategies for sampling and learning in the presence of multiple reward functions of different fidelities, and we provide a number of new concrete multi-fidelity algorithms under this

Submitted to 37th Conference on Neural Information Processing Systems (NeurIPS 2023). Do not distribute.

framework. We present empirical evaluations in Symbolic Regression and Antibody Optimization, where the former is used for benchmarking and comparing algorithms and the latter is our application of interest to find effective antibody sequences for binding SARS-CoV-2 viruses.

## 2   Background

A symbolic optimization problem is specified by a library $\mathcal{L}$ and a reward function $R$. The library $\mathcal{L} = \{\tau^1, \ldots, \tau^t\}$ is a set of *tokens* $\tau^i$ that determine the space $\mathbb{T}$ of possible *token sequences* $\tau = (\tau_1, \ldots, \tau_n)$, $n \leq n_{\max}$, that are candidate solutions to the SO problem. The reward function $R : \mathbb{T} \to \mathbb{R} \cup \{-\infty\}$ evaluates the fitness of each sequence; invalid sequences are assigned value $-\infty$. The main challenge of SO is to search within $\mathbb{T}$, which scales exponentially with the maximum sequence length $n_{\max}$, for the sequence or set of sequences that maximizes the reward:

$$\underset{n \in \mathbb{N}, \tau \in \mathbb{T}}{\arg\max} \left[ R(\tau) \right] \text{ with } \tau = (\tau_1, \ldots, \tau_n), \tau_i \in \mathcal{L}. \tag{1}$$

Our novel approach for multi-fidelity symbolic optimization builds on *Deep Symbolic Optimization* (DSO) (Petersen et al., 2021), as it is the current state-of-the-art in general purpose, real-world, SO domains (Landajuela et al., 2021; Silva et al., 2022). DSO uses a recurrent neural network policy to construct a token sequence autoregressively by sampling each token (action) conditioned on the sequence of previous tokens generated so far (observation). To optimize for the discovery of the *best* token sequence, DSO maximizes the best-case performance using the Risk-Seeking Policy Gradient

$$J(\theta) := \mathbb{E}_\theta \left[ R(\tau) \mid R(\tau) \geq Q_\epsilon \right], \tag{2}$$

$$\nabla_\theta J \approx \frac{1}{\epsilon N} \sum_{i=1}^{N} \left[ R(\tau^{(i)}) - \tilde{R}_\epsilon(\theta) \right] \cdot \mathbf{1}_{R(\tau^{(i)}) \geq \tilde{R}_\epsilon(\theta)} \nabla_\theta \log p(\tau^{(i)}|\theta), \tag{3}$$

where $Q_\epsilon$ is the $(1 - \epsilon)$-quantile of the reward distribution under the policy, $\tilde{R}_\epsilon(\theta)$ is the empirical $(1 - \epsilon)$-quantile of the batch of rewards, and $\mathbf{1}_x$ returns 1 if condition $x$ is true and 0 otherwise.

## 3   Problem Statement

We introduce the Multi-Fidelity Markov Decision Process (MF-MDP) as the tuple $\left( S, A, \boldsymbol{T}^{MF}, \boldsymbol{R}^{MF} \right)$. As in regular MDPs, $S$ is the state space and $A$ is the action space. However, MF-MDPs have multiple state transition and reward functions:

$$\boldsymbol{T}^{MF} = \langle T^0, T^1, \ldots, T^{f_{\max}} \rangle \qquad \boldsymbol{R}^{MF} = \langle R^0, R^1, \ldots, R^{f_{\max}} \rangle,$$

where $f_{\max}+1$ is the number of fidelities. Each pair of transition and reward functions $\Xi^f := (T^f, R^f)$ determines a unique *source* environment for fidelity $f \in \{0, \ldots, f_{\max}\}$. We assume all fidelities share the same state-action space. Fidelities can be freely chosen at the start of each finite episode and persist until episode termination. The fidelity $f = 0$ ($\Xi^0$) is the "real environment", and therefore the agent objective is to maximize the reward in this fidelity. Since we are here concerned with SO problems, for which the rewards are computed only for whole token sequences, we are dealing with a sub-class of MF-MDPs where: (i) the transition function is the same for all fidelities; (ii) the reward is evaluated only for complete trajectories (token sequences). Hence, for this paper the optimal solution can be described as:

$$\underset{n \in \mathbb{N}, \tau}{\arg\max} \left[ R^0(\tau) \right] \text{ with } \tau = (\tau_1, \ldots, \tau_n), \tau_i \in \mathcal{L}, \tag{4}$$

where selecting each token $\tau_i$ is an agent action. However, each source $\Xi^f$ is associated to a sampling cost $c$, where $\forall f > 0 : c(\Xi^0) \gg c(\Xi^f)$, such that solely relying on $\Xi^0$ is infeasible, whereas the cost for non-zero fidelities is negligible[1] relative to the cost for $\Xi^0$. However, all other sources $\Xi^f$ approximate the real-world source $\Xi^0$, and can be used to bootstrap learning and enable finding a good policy for $\Xi^0$ with a reduced number of samples.

---

[1]This holds for many real-world applications, e.g., using a machine learning model approximating antibody binding affinity is orders of magnitude cheaper than carrying out wet lab experiments.

75 MF-MDPs can be used to model a wide variety of domains in both general RL and SO, such as
76 robotics (Kober et al., 2013) and antibody design (Norman et al., 2020), where evaluating solutions is
77 expensive or dangerous but simulations are available.

# 4 Solving Multi-fidelity Symbolic Optimization

79 We propose a general framework called *Multi-Fidelity Deep Symbolic Optimization* (MF-DSO) that
80 iterates between two steps: sampling and learning. MF-DSO encapsulates a large set of possible
81 concrete algorithms as different multi-fidelity strategies may be designed for each stage.

82 1. **Multi-fidelity Sampling**: At the start of each episode, a `SAMPLE` method chooses a fidelity, or
83    multiple fidelities, and executes the corresponding transition and reward functions for that episode.
84    Since only the reward function (not the transition function) changes between fidelities in SO, we
85    generate a batch $\mathcal{T}$ of trajectories (token sequences) using the current policy and choose a fidelity
86    or multiple fidelities for each trajectory.

87 2. **Multi-fidelity Learning**: Given a batch of sampled trajectories $\mathcal{T}$, which may contain a mixture of
88    samples with rewards calculated using different fidelities, a multi-fidelity learning strategy `LEARN`
89    takes a policy update step to learn how to sample better token sequences (aiming at Equation (1)).

90 Algorithm 1 describes the training loop, which iterates between `SAMPLE` and `LEARN` until a termination
91 condition is achieved (for example, a budget number of samples in $f = 0$ or a total wall-clock run
92 time). During the course of training, we save the *Hall of Fame* (`HoF`), defined as the set of best
93 samples (according to the best fidelity seen at the moment) found so far.

---

**Algorithm 1** Multi-Fidelity Deep Symbolic Optimization

---

**Require:** $\pi_\theta$: Policy network parameterized by $\theta$; $\Xi$: set of available fidelities; $n_b$: Size of the
batch; `SAMPLE`: method for defining which fidelity to use; `LEARN`: method for updating the policy
network.
1: `HoF` $\leftarrow \emptyset$
2: initiate network parameters $\theta$
3: **while** termination condition not achieved **do**
4:     $\mathcal{T} \leftarrow \pi_\theta(n_b)$                                                   ▷ Generate samples
5:     $\{\text{fid}(\tau)\} \leftarrow$ `SAMPLE`$(\mathcal{T}, \Xi)$                              ▷ Fidelity for each sample
6:     **for** $\forall \tau \in \mathcal{T}$ **do**
7:         $f \leftarrow \min(\text{fid}(\tau.r), \text{fid}(\tau))$                              ▷ Define best fidelity
8:         $\tau.r \leftarrow R^f(\tau)$                                        ▷ Define reward according to chosen fidelity
9:     **end for**
10:     $\theta \leftarrow$ `LEARN`$(\theta, \mathcal{T})$                                        ▷ Update Policy Network
11:     `HoF` $\leftarrow hof\_update($`HoF`$, \mathcal{T})$
12: **end while**
13: **return** `HoF`

---

## 4.1 Multi-fidelity Sampling

95 The sampling algorithm is critical to a multi-fidelity problem, given that a high cost is spent every
96 time a sample is evaluated in fidelity $f = 0$. Hereafter, let $\text{fid}(\cdot)$ denote the fidelity of the argument,
97 and we use the "object.property" notation $\tau.r$ to denote the highest fidelity reward of $\tau$ evaluated so
98 far— e.g. $\text{fid}(\tau.r)$ is the highest fidelity of $\tau$ seen so far.

99 Our proposed sampling method is called **Elite-Pick sampling** and is based on an elitism mechanism.
100 Similar to Cutler et al. (2014), we follow the intuition that it is advantageous to sample in the
101 highest fidelity only when we expect to have a high performance, so that computational budget is
102 concentrated on promising solutions. Initially, all samples are initially evaluated in a low (non-zero)
103 fidelity: $\tau.r = R^{\text{low}}(\tau)$. Thereafter, each `SAMPLE` step uses $\tau.r$ to calculate the empirical $(1 - \rho)$-
104 quantile $Q_\rho(\mathcal{T})$, where $\rho \in (0, 1)$ is a fixed threshold, and only the samples in the top quantile (i.e.,
105 those with $\tau.r > Q_\rho$) are evaluated in $R^0$:

$$\text{SAMPLE}_\rho = \begin{cases} 0 & \text{if } \tau.r \geq Q_\rho \\ \text{low} & \text{otherwise} \end{cases} : \tau \in \mathcal{T} \tag{5}$$

When more than two fidelities are available, one may interpret *low* by randomly sampling from a mixture of the non-zero fidelities.

## 4.2 Multi-fidelity Learning

After each sample is assigned a fidelity, the learning algorithm uses $\mathcal{T}$ to update the policy network. We propose two learning algorithms that explictly account for the fact that $\mathcal{T}$ may contain a mixture of samples in different fidelities.

**Weighted Policy Gradient**: This is a policy gradient algorithm where the highest fidelity $f = 0$ receives weight $\gamma$ and all other fidelities receive weight $1 - \gamma$, for $\gamma \in (0, 1)$.

$$J_{PG}(\theta) := \mathbb{E}\left[\gamma l(R^0(\mathcal{T}_0)) + \sum_{f=1}^{f=f_{\max}} \frac{1 - \gamma}{f_{\max} - 1} l(R^f(\mathcal{T}_f))\right] \tag{6}$$

$l$ is a simple loss function, which we chose to be REINFORCE (Williams, 1992) in this work. This learning algorithm with elite-pick sampling is henceforth called **PGEP**. We also consider a variation of the algorithm where all fidelities have the same weight **PGEP_u**.

Following PGEP, we introduce a more principled algorithm for this problem.

**Multi-Fidelity Risk-Seeking**: Inspired by the risk-seeking policy gradient algorithm described in Section 2, we propose a multi-fidelity risk-seeking objective:

$$J_\epsilon(\theta) := \mathbb{E}_\theta\left[R^0(\tau) \mid \tau.r \geq Q_\epsilon^m\right], \tag{7}$$

where $Q_\epsilon^m$ is the top $(1 - \epsilon)$-quantile of $\tau.r$ for $\tau \in \mathcal{T}$. Here, and in the following, we use superscript "m" to denote "mixture", since $Q_\epsilon^m$ is computed on a batch of samples whose $\tau.r$ may belong to different fidelities. Intuitively, we want to find a distribution such that the top $\epsilon$ fraction of samples (as evaluated using the best fidelity sampled so far) have maximum performance when evaluated using the highest fidelity reward $R^0$. Crucially, note that (7) is well-defined at each iteration of the algorithm based on the fidelity of $\tau.r$ for each $\tau \in \mathcal{T}$, while the fidelities may improve after an iteration when a better fidelity is sampled. We can find a local maximum of (7) by stochastic gradient ascent along (9) in the following result, where the full proof is given in Appendix B.

**Proposition 1.** *Let random variable $\tau$ have distribution $\pi_\theta$, and let $R^0$ and $R^m$ be two functions of $\tau$ with induced distributions $p^0$ and $p^m$. Let $F_\theta^m$ denote the CDF of $p^m$. Let $Q_\epsilon^m(\theta) = \inf_{\tau \in \Omega}\{R^m(\tau) : F_\theta^m(r) \geq 1 - \epsilon\}$ denote the $(1 - \epsilon)$-quantile of $p^m$. The gradient of*

$$J_\epsilon(\theta) := \mathbb{E}_\theta\left[R^0(\tau) \mid R^m(\tau) \geq Q_\epsilon^m(\theta)\right] \tag{8}$$

*is given by*

$$\nabla_\theta J_\epsilon(\theta) = \mathbb{E}_\theta\left[\nabla_\theta \log \pi_\theta(\tau)(R^0(\tau) - R^0(\tau_\epsilon)) \mid R^m(\tau) \geq Q_\epsilon^m(\theta)\right] \tag{9}$$

*where $\tau_\epsilon = \arg\inf\{R^m(\tau) : F_\theta^m(r) \geq 1 - \epsilon\}$ is the sample that attains the quantile.*

We call Risk-Seeking learning allied with Elite-Pick sample generation as **RSEP** henceforth. We also consider a variation of the algorithm where, after sampling is applied, all the samples currently sampled in $f = 0$ are used to recalculate $Q_{\epsilon_2}^{\text{fid}(\tau.r)=0}$, filtering out samples with lower rewards than the $\epsilon_2$-quantile, which can further discard low-quality samples in $f = 0$. This means that additional samples might be discarded from the learning update based on their updated value of $\tau.r$ after the sampling process. We name this variation as **RSEP_0**.

### 4.2.1 Theoretical Analysis of RSEP

Since RSEP uses a mixture of low- and high-fidelity samples to compute the quantile for filtering (i.e. selecting $\tau : \tau.r \geq Q_\epsilon^m$), whereas one would use only $Q_\epsilon^0$ if this were feasible, we would like to understand the probability of *wrong exclusion*: the probability that a sample would have passed the high-fidelity filter $Q_\epsilon^0$ but was wrongly rejected by the low-fidelity filter $Q_\epsilon^m$. Assuming that the error between the highest fidelity and other fidelities can be modeled by a distribution, Proposition 2 below, derived in Appendix B, states that the probability of wrong exclusion scales according to the cumulative distribution of the error as a function of the difference in quantiles.

**Proposition 2.** *Let $R^0$ and $R^1$ be random variables related by error distribution $N$: $R^1 := R^0 + N$. Let $Q^0_\epsilon$ and $Q^1_\epsilon$ be the $(1 - \epsilon)$-quantiles of the distributions of $R^0$ and $R^1$, respectively. Then*

$$P(R^0 \geq Q^0_\epsilon, R^1 \leq Q^1_\epsilon) = \epsilon \mathbb{E}\left[ F_N(Q^1_\epsilon - R^0) \mid R^0 \geq Q^0_\epsilon \right] \tag{10}$$

*where $F_N(r)$ is the CDF of the error distribution.*

From this, we can make two intuitive observations: 1) the smaller the $\epsilon$ used by the "true" high-fidelity risk-seeking objective, the smaller the probability of error; 2) The smaller the low-fidelity quantile $Q^1_\epsilon$, the more likely a sample is to pass the low-fidelity filter, and the smaller the probability of error.

Furthermore, we show in the following that the RSEP algorithm eventually maximizes the same objective as the risk-seeking policy gradient. First, we need the following assumption:

**Assumption 1.** *One of the following holds:*

- *Case 1: As part of the sampling method, we include a non-zero probability of sampling $f = 0$ for each trajectory $\tau$ regardless of its current $\tau.r$.*

- *Case 2: For all $\tau \in \mathcal{T}$, we have $R^0(\tau) \leq R^f(\tau)$ for $f \neq 0$.*

Case 2 arises in real-world scenarios where lower-resolution fidelities $f \neq 0$ are overly optimistic—e.g., a robotics simulation that does not penalize actions that would cause real-world mechanical damage. Intuitively, this condition avoids the problematic scenario where a sample with high $R^0$ was wrongly filtered out due to a bad lower-fidelity estimate.

**Proposition 3.** *Let $J_{risk}$ be the Risk-Seeking Policy Gradient objective:*

$$J_{risk}(\theta) := \mathbb{E}_\theta\left[ R^0(\tau) \mid R^0(\tau) \geq Q^0_\epsilon \right] \tag{11}$$

*and $J_{RSEP}$ be the RSEP objective (Equation 7). Given Assumption 1, optimizing for the RSEP objective, in the limit of infinite exploration, corresponds to optimizing for the risk-seeking objective.*

However, we expect that high-quality sequences will be found much quicker than when using a single fidelity, which will be shown in the empirical evaluation.

## 5 Empirical Evaluation

We empirically evaluate our methods for MF-DSO in two real-world problems, *Symbolic Regression* and *Antibody Optimization*. While both domains are of practical importance, the former provides well-defined benchmarks and experiments can be performed quickly, while the latter represents a challenging domain where freely sampling in the highest fidelity is infeasible.

### 5.1 Symbolic Regression

Symbolic regression is the problem of searching over a space of tractable (i.e. concise, closed-form) mathematical expressions that best fit a set of observations. This can be used, for example, to discover equations that explain physical phenomena. Specifically, given a dataset $(\boldsymbol{X}, \boldsymbol{y})$, where each observation $\boldsymbol{X_i} \in \mathbb{R}^n$ is related to a target value $y_i \in \mathbb{R}$ for $i = 1, \ldots, m$, symbolic regression aims to identify a function $f : \mathbb{R}^n \to \mathbb{R}$, in the form of a short mathematical expression, that best fits the dataset according to a measure such as mean squared error. Symbolic regression is useful for our preliminary evaluation of multi-fidelity symbolic optimization methods because: (i) challenging benchmarks (White et al., 2013) and strong baseline methods (Schmidt and Lipson, 2009) are well-established ; (ii) success criteria is clearly defined in problems with ground-truth expressions; and (iii) computing the quality of candidate expressions is easy, allowing repeated experiments to achieve statistically significant results.

We leverage the Nguyen symbolic regression benchmark suite (Uy et al., 2011), a set of 12 commonly used expressions developed and vetted by the symbolic regression community (White et al., 2013). We use the ground truth expression to generate training and test data, and we define the highest fidelity reward for a candidate function $f_\tau$—represented by a token sequence $\tau$ that is the pre-order traversal of the expression tree of the function—based on its error from the targets $\boldsymbol{y}$: $R^0(\tau, \boldsymbol{X}) = 1 - \sqrt{\frac{1}{m} \sum_{i=1}^m (f_\tau(\boldsymbol{X}_i) - y_i)^2}$. We define lower-fidelity rewards using $(\boldsymbol{X}, \boldsymbol{y})$ as follows.

1. $\Xi^1$: We add white noise to the targets $y$, so that the rewards are calculated using $(X, y + \epsilon)$.

2. $\Xi^2$: We train a simple Gaussian Process Regressor $m$ on the data, and use $m(X)$ instead of $y$. This simulates a common situation where a surrogate model is trained in real-world data to provide a faster and cheaper low-fidelity estimator.

We show results for Nguyen 4-6, 9, 10, and 12 for all experiments in the main text of the paper. Those benchmarks were chosen because they represent the main trend of the results for both middle- and high-difficulty ground truth equations. The results for all 12 benchmarks, as well as the full description of their ground truth equations, are shown in the supplemental material.

### 5.1.1 Baseline Multi-Fidelity Performance

This series of experiments aim to answer the question *"Is it useful to use multi-fidelity samples?"*; and to assess the performance of simple multi-fidelity strategies. The following baselines are considered:

- **Upper bound**: Only uses $\Xi^0$. Given unlimited samples, this baseline should be the top performer. However, we are here interested in the scenario in which samples from the highest fidelity are limited.
- **Lower bound**: Only uses $\Xi^1$ and $\Xi^2$. This baseline shows the agent performance in the lower fidelities when the real world is not available.
- **Sequential**: This is a transfer learning approach, whereby learning is carried out in $\Xi^1$ and $\Xi^2$ for a number of iterations, before switching to solely using $\Xi^0$ until termination.
- **Shuffled**: This baseline randomly samples from different fidelities according to a fixed probability. The highest fidelity is set to around $9\%$ of probability to be sampled from.

Figure 1 shows the best sample found per number of explored samples in $f = 0$. Although those graphs cannot be interpreted alone[2], they present a gross estimation of the learning speed of each algorithm. Table 1 shows the average quality of the hall of fame after training, providing the extra information we needed to assess the performance. As expected, *lower bound* shows that sampling only from the lowest fidelity produces solutions that perform poorly in the highest fidelity. Although *shuffled* sometimes achieves high-performing samples (e.g., on Nguyen-6), the mixture of fidelities produces inconsistent reward signal for the same sample, which results in low *avg* metric overall in most of the benchmarks. Despite the poor performance from those aforementioned baselines, *sequential* reveals the the benefit of using lower fidelities to bootstrap learning, as it consistently achieves better performance than *upper bound* with the same budget of samples from $f = 0$ (evidenced in both Figure 1 and Table 1). Having established the advantage of using multiple fidelities, our next experiment will show that MF-DSO outperforms the baselines.

Table 1: The results represent the best (max) and the average (avg) quality of samples in the hall of fame by the end of the training process. Averages across 150 repetitions.

| Benchmark | Lower bound | | Upper bound | | Shuffled | | Sequential | |
|---|---|---|---|---|---|---|---|---|
| | Max | Avg | Max | Avg | Max | Avg | Max | Avg |
| Nguyen-4 | 0.884 | 0.703 | 0.890 | 0.788 | 0.923 | 0.786 | **0.925** | **0.846** |
| Nguyen-5 | 0.530 | 0.257 | 0.705 | 0.511 | 0.728 | 0.505 | **0.754** | **0.563** |
| Nguyen-6 | 0.800 | 0.578 | 0.918 | 0.820 | 0.966 | 0.839 | **0.969** | **0.859** |
| Nguyen-9 | 0.396 | 0.300 | 0.832 | 0.702 | 0.875 | 0.720 | **0.889** | **0.761** |
| Nguyen-10 | 0.498 | 0.355 | 0.851 | 0.726 | 0.872 | 0.712 | **0.883** | **0.744** |
| Nguyen-12 | 0.526 | 0.366 | 0.706 | 0.561 | 0.758 | 0.505 | **0.777** | **0.621** |

### 5.2 Symbolic Regression Evaluation

We evaluate the performance of all MF-DSO methods proposed above: *RSEP*, *RSEP_0*, *PGEP*, and *PGEP_u*; as well as the best performing baseline method *sequential*. Figure 2 and Table 2 show

---

[2]A good sample found by the algorithm is not necessarily stored in the hall of fame. If a sample is overestimated in another fidelity, the best sample so far might be discarded depending on the algorithm used (this happens, e.g., with *shuffled*)

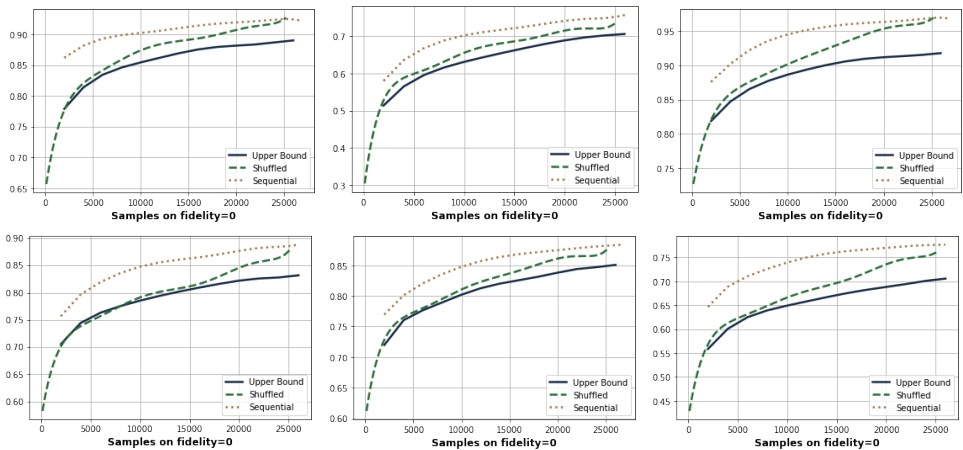

Figure 1: Average reward of best sample found so far during training (x-axis is the amount of samples from $\Xi^0$) across 150 repetitions. Nguyen 4-6, 9, 10, 12 are depicted from left to right, top to bottom. Curves are polynomial interpolation of each experiment curve for ease of visualization.

the results for all benchmarks. For both the *max* and *avg* metrics, *RSEP* outperformed all other algorithms in, respectively, 4 and 2 of the benchmarks, which clearly makes it the best performing algorithm in this experiment. *RSEP_0*, a variant of the same algorithm, ranked best of all algorithms in 1 and 2 of benchmarks for each of the metrics. Finally, *PGEP_u* ranked best 1 and 2 times in the metrics. Notably, the best baseline method (sequential) was not able to outperform the multi-fidelity-specific algorithms in any of the metrics or benchmarks. We conclude from this experiment that the proposed algorithms provide significant gains in multi-fidelity environments, and the overall strong performance of *RSEP* and *PGEP_u* motivates us to use them in the more computationally expensive experiments in the next section.

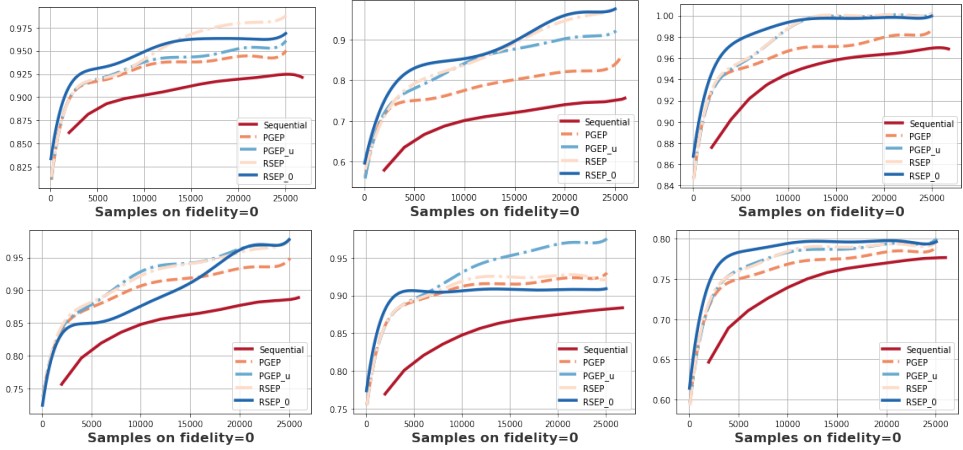

Figure 2: Average reward of best sample found so far during training (x-axis is the amount of samples from $\Xi^0$) across 150 repetitions. Nguyen 4-6, 9, 10, and 12 are depicted from left to right.

## 5.3 Antibody Optimization

Antibodies are proteins—sequences of amino acids with target-specific complementarity determining regions (CDRs) (Wu and Kabat, 1970)—that serve as the human immune system's primary line of defense by binding to, and neutralizing, harmful antigens (e.g., a virus or bacteria). They can be manufactured and directly administered to patients (Carter, 2006), but the design of efficacious antibodies (Norman et al., 2020) is still a challenge as the set of 20 canonical amino acids define a search space of $20^L$, where $L$ is the number of sequence positions to be optimized. Exhaustively

Table 2: The results represent the best (max) and the average (avg) quality of samples in the hall of fame by the end of the training process. Averages across 150 repetitions. Best results for each metric are highlighted in bold.

| Benchmark | Sequential | | PGEP | | PGEP_u | | RSEP | | RSEP_0 | |
|---|---|---|---|---|---|---|---|---|---|---|
| | Max | Avg | Max | Avg | Max | Avg | Max | Avg | Max | Avg |
| Nguyen-4 | 0.925 | 0.846 | 0.946 | 0.894 | 0.956 | 0.921 | 0.985 | 0.947 | **0.991** | **0.961** |
| Nguyen-5 | 0.754 | 0.563 | 0.832 | 0.668 | 0.913 | 0.761 | **0.966** | 0.801 | 0.960 | **0.823** |
| Nguyen-6 | 0.969 | 0.859 | 0.983 | 0.944 | 0.999 | **0.981** | **1.000** | 0.965 | 0.999 | 0.942 |
| Nguyen-9 | 0.889 | 0.761 | 0.941 | 0.838 | 0.972 | 0.849 | **0.973** | **0.858** | 0.968 | 0.844 |
| Nguyen-10 | 0.883 | 0.744 | 0.925 | 0.858 | **0.971** | **0.901** | 0.927 | 0.862 | 0.908 | 0.819 |
| Nguyen-12 | 0.777 | 0.621 | 0.786 | 0.691 | 0.796 | 0.762 | **0.792** | **0.776** | 0.795 | 0.772 |

searching this space is infeasible due to the high cost of performing wet lab experimentation of antibodies, or even the high computational cost of running simulations. We follow a *rapid response* approach. We start with an existing *parental* antibody that is effective against a known antigen (e.g. SARS-CoV-1) in the same viral family as the target (e.g., SARS-CoV-2) , but which does not bind effectively to the new target. Given that both antigens are related, the symbolic optimization problem is to construct a set of *multi-point mutations* in the CDR of the existing antibody to improve binding to the new target, rather than randomly exploring the space of possible antibodies. For the experiments in this paper we have chosen the BQ.1.1 Omicron variant of the SARS-CoV-2 virus to be our target (Hefnawy et al., 2023).

We define reward functions with two fidelities using the opposite of the $ddG^3$ computed, *in silico*, using the Rosetta Flex (Barlow et al., 2018) simulation of antibody-antigen interactions.

- $\Xi^0$: The highest fidelity reward $R^0$ uses a Rosetta Flex simulation of the full multi-point mutation to compute an accurate binding score. In this way, Rosetta Flex utilizes samples of protein conformation to generate an ensemble of structures and averages their constituent ddG values. Although this provides a good estimation the reward, computing a single reward value in this way takes approximately 15 hours of a CPU on our high-performance computing platform described below.

- $\Xi^1$: For a quick estimation of our ddG rewards, before starting the learning process, we run a single Rosetta Flex simulation for every single possible individual mutation in the interface between the parental antibody and antigen[4]. After we have an estimate ddG for each possible single-point mutation, we estimate the ddG for our candidate antibody as the sum of all single-point mutation differences. This measure is correlated to $R^0$ but not as precise, as we show in Appendix D.

Given the long time needed to run experiments in this domain, we leverage the results from the Symbolic Regression experiments to down-select algorithms for the evaluation in this domain. We evaluate: (i) **Upper Bound**: training only on $\Xi^0$; (ii) **Sequential**: training on $\Xi^1$ for a long time then switching to $\Xi^0$; (iii) **RSEP**; and (iv) **PGEP**, the top-performing algorithms from each category in the last section. Each algorithm was given the same budget of 250 hours to run the experiment, where all algorithms were executed in parallel with 72 Intel Xeon E5-2695 v2 CPUs each. As running full Rosetta simulations is significantly slower than any other operation, this results in an approximately equivalent high-fidelity budget for all algorithms.

## 5.4 Antibody Optimization Evaluation

The results for the antibody optimization domain are shown in Table 3 and Figure 3. For this domain, it is more important to have a *set* of promising antibodies than a single one because other aspects apart from the binding score (such as stability, safety, and manufacturability) have to be taken into account when the antibodies are evaluated in the wet lab. Hence having a set of candidate antibodies increase the probability of finding a candidate good in all properties and thus the figure (right) show

---

[3]ddG, or $\Delta\Delta G$, is the change in Gibbs free energy upon mutation. A lower value mean that the new antibody binds to the target better than the parental antibody.

[4]Notice that this requires running a simulation only for the 20 possible amino acids for every possible position, which is a much smaller space than the combinatorial space of multiple simultaneous mutations required for $\Xi^0$.

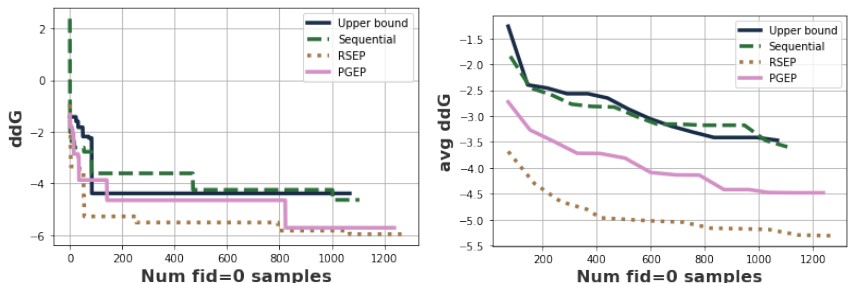

Figure 3: Best ddG found (left) and average of the ddG in the hall of fame (right) so far during training. Lower ddG is better.

the average ddG between the 10 best antibodies during training. The results for this domain are slightly different from our initial results from the symbolic regression domain. Although *upper bound* initiates with a much worse initial batch of samples, it quickly overtakes *Sequential*, finishing almost tied up with thus baseline. This trend, that is different from what was observed in the first evaluation domain, shows that those algorithms were more deeply affected by the difference in fidelities than in the symbolic regression domain where we fabricated noise. This makes sense given *Sequential* was trained initially in the low fidelity and can suffer from negative transfer. On the other hand, our multifidelity-aware algorithms *PGEP* and *RSEP* outperformed the baselines by a large margin since the first batch (more visible in the right figure), showing the flexibility and power of our algorithms. The table clearly shows that RSEP outperforms the other algorithms in both metrics, improving the average significantly over the baselines and *PGEP*. Those empirical results are very encouraging for our multifidelity algorithms, showing in a very complex and relevant application that it is worthy to explicitly reason over multiple fidelities in symbolic optimization tasks.

Table 3: Best and Average ddG score for the antibodies in the hall of fame at the end of the training process. Best results are in bold.

| Alg | *Best* | *Avg* |
|---|---|---|
| **Upper Bound** | -4.38 | -3.46 |
| **Sequential** | -4.64 | -3.59 |
| **RSEP** | **-5.96** | **-5.31** |
| **PGEP** | -5.72 | -4.48 |

## 6 Conclusion and Further Work

Although many applications are naturally modeled as multi-fidelity problems, the literature has predominantly coped with those environment in an ad-hoc manner. Those problems are either modeled as Transfer Learning problems or as a simulation-optimization problem where the low-fidelity environment is iteratively refined but the learning algorithm is unaware of the multiple fidelities. We propose to explicitly reason over the multiple fidelities and leverage lower fidelity estimates to bias the sampling in the higher, more expensive, fidelity. We contribute the description of Multi-Fidelity MDPs (MF-MDPs), defining a new challenge to the community. We also contribute two families of algorithms for MF-MDPs specialized for SO problems: *RSEP* and *PGEP*. Moreover, we perform an empirical evaluation in the Symbolic Regression and Antibody Optimization domains, showing that MF-MDP-based algorithms outperform baseline strategies in both domains. The conclusion of our experimentation is that *RSEP* is the best performing algorithm overall and should be the first choice, but since *PGEP* was the best performer in some of the symbolic regression benchmarks, it is worthy to also evaluate it in cases where this is feasible. Further work includes explicitly reasoning over the cost of sampling from each fidelity, instead of assuming that samples are free from all lower fidelities as we do in this paper. Another avenue is proposing algorithms that work for a broader class of MF-MDPs, solving more applications of interest, including sim2real RL domains.

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

407   *Representations (ICLR)*.

## A  Additional Related Works

Although MF-MDPs have not been formally described before, multi-fidelity rewards have already been explored in the literature (Beran et al., 2020; Peherstorfer et al., 2018). Even though the agent end goal is to optimize performance in the fidelity 0, a group of works propose ways to iteratively finetune lower-fidelity surrogate models to make them more realistic and enable training directly in the lower, cheaper to sample from, fidelity. A common way to handle the multiple fidelities is either through modifying lower-fidelity transition (Hanna et al., 2021; Christiano et al., 2016; Golemo et al., 2018; Abbeel et al., 2006) or reward (Iocchi et al., 2007) functions, by learning a correction factor that approximates them to the highest fidelity function. We, on the other hand, focus on explicitly using both lower and higher-fidelity estimates to learn, instead of fine tuning the lower fidelity models.

The multi-fidelity problem has been explored in an ad hoc manner as a Transfer Learning problem (Silva and Costa, 2019), where the lower fidelity is solved, and the solution is somehow reused to learn in the highest fidelity (Aydin et al., 2019). This approach is mimicked by our *sequential* baseline and, as shown in our experiments, is not as effective and explicitly reasoning over the multiple fidelities during learning.

Some Neural Architecture Search works considered this application as a multi-fidelity problem (Trofimov et al., 2020; Yang et al., 2022), because each candidate architecture can be evaluated for an arbitrarily number of training iterations, resulting in an as higher fidelity reward as longer you train the model. However, the key distinction from our method is that, in their modeling, for evaluating a sample in a given fidelity, the rewards for all lower fidelities *must* be computed, which is not the case in our modeling and applications.

Perhaps most similar to our paper are the works from Khairy and Balaprakash (2022) and Cutler et al. (2014). In the former, they consider that the state space of the low-fidelity environment is an abstracted version of the high-fidelity one (and therefore smaller). We instead assume that the state space is the same and the lower fidelity simply uses a cheaper approximate way of calculating the reward. In the latter, the authors assume that the agent can estimate its epistemic uncertainty and only queries the high fidelity when the uncertainty is low, so as to avoid exploring low-quality samples in the high fidelity. While our method similarly try to bias the evaluations in the highest fidelity towards high-performing samples, we do not require uncertainty calculation, which might be difficult to do.

Outside of the RL/SO communities, several works constrain optimization within a trust-region when using lower-fidelity estimates (Robinson et al., 2006). While those methods are not directly-usable in RL or SO problems, it might inspire TRPO-like (Schulman et al., 2015) methods using our formulation.

## B  Proofs

**Proposition 1.** *Let random variable $\tau$ have distribution $\pi_\theta$, and let $R^0$ and $R^m$ be two functions of $\tau$ with induced distributions $p^0$ and $p^m$. Let $F_\theta^m$ denote the CDF of $p^m$. Let $Q_\epsilon^m(\theta) = \inf_{\tau \in \Omega} \{R^m(\tau) \colon F_\theta^m(r) \geq 1 - \epsilon\}$ denote the $(1 - \epsilon)$-quantile of $p^m$. The gradient of*

$$J_\epsilon(\theta) := \mathbb{E}_\theta \left[ R^0(\tau) \mid R^m(\tau) \geq Q_\epsilon^m(\theta) \right] \tag{8}$$

*is given by*

$$\nabla_\theta J_\epsilon(\theta) = \mathbb{E}_\theta \left[ \nabla_\theta \log \pi_\theta(\tau)(R^0(\tau) - R^0(\tau_\epsilon)) \mid R^m(\tau) \geq Q_\epsilon^m(\theta) \right] \tag{9}$$

*where $\tau_\epsilon = \arg\inf\{R^m(\tau) \colon F_\theta^m(r) \geq 1 - \epsilon\}$ is the sample that attains the quantile.*

*Proof.* First, we provide an elementary proof for the case where $\tau$ is a scalar random variable, then we provide a proof for the multi-dimensional case.

**Single-dimensional case.** Define the set of samples for which the mixture reward exceeds the $1 - \epsilon$ quantile:

$$D_\theta := \{\tau \in \Omega \colon R^m(\tau) \geq Q_\epsilon^m(\theta)\} \tag{12}$$

We expand the definition of the objective:

$$J_\epsilon(\theta) = \int_\Omega R^0(\tau) f_{\theta, R^m(\tau) \geq Q_\epsilon^m(\theta)}(\tau) d\tau \tag{13}$$

$$= \int_\Omega R^0(\tau) \frac{f_\theta(\tau, R^m(\tau) \geq Q_\epsilon^m(\theta))}{f_\theta(R^m(\tau) \geq Q_\epsilon^m(\theta))} d\tau \tag{14}$$

$$= \frac{1}{\epsilon} \int_\Omega R^0(\tau) f_\theta(\tau, R^m(\tau) \geq Q_\epsilon^m(\theta)) d\tau \tag{15}$$

$$= \frac{1}{\epsilon} \int_{\tau \in D_\theta} R^0(\tau) \pi_\theta(\tau) d\tau \tag{16}$$

Assuming sufficient continuity of the reward, policy, and quantile as a function of parameter $\theta$, we can apply the Leibniz integral rule to differentiate under the integral sign. Differentiating both sides of

$$\epsilon = \int_{\tau \in D_\theta} \pi_\theta(\tau) d\tau, \tag{17}$$

and letting $b$ denote the upper bound of the reward, we have

$$0 = \nabla_\theta \int_{\tau \in D_\theta} \pi_\theta(\tau) d\tau \tag{18}$$

$$= \nabla_\theta \int_{R^m(\tau_\epsilon(\theta))}^b p_\theta^m(r) dr \tag{19}$$

$$= -p_\theta^m(R^m(\tau_\epsilon)) \nabla_\theta R^m(\tau_\epsilon(\theta)) + \int_{R^m(\tau_\epsilon(\theta))}^b \nabla_\theta p_\theta^m(r) dr \tag{20}$$

Let $\tau_r$ denote the sample that satisfies $R^m(\tau) = r$. Note in particular that $R^m(\tau_{R^m(\tau_\epsilon)}) = R^m(\tau_\epsilon)$, so that $\tau_{R^m(\tau_\epsilon)} = \tau_\epsilon$. Using this fact and applying the Leibniz integral rule to the objective (16), we have

$$\nabla_\theta J_\epsilon(\theta) = \nabla_\theta \frac{1}{\epsilon} \int_{R^m(\tau_\epsilon(\theta))}^b R^0(\tau_r) p_\theta^m(r) dr \tag{21}$$

$$= -\frac{1}{\epsilon} R^0(\tau_\epsilon(\theta)) p_\theta^m(R^m(\tau_\epsilon(\theta))) \nabla_\theta R^m(\tau_\epsilon(\theta)) \tag{22}$$

$$+ \frac{1}{\epsilon} \int_{R^m(\tau_\epsilon(\theta))}^b R^0(\tau_r) \nabla_\theta p_\theta^m(r) dr$$

Substituting eq. (20) into eq. (22), we get

$$\nabla_\theta J_\epsilon(\theta) = -\frac{1}{\epsilon} R^0(\tau_\epsilon(\theta)) \int_{R^m(\tau_\epsilon(\theta))}^b \nabla_\theta p_\theta^m(r) dr \tag{23}$$

$$+ \frac{1}{\epsilon} \int_{R^m(\tau_\epsilon(\theta))}^b R^0(\tau_r) \nabla_\theta p_\theta^m(r) dr$$

$$= \frac{1}{\epsilon} \int_{R^m(\tau_\epsilon(\theta))}^b \nabla_\theta p_\theta^m(r) \left( R^0(\tau_r) - R^0(\tau_\epsilon(\theta)) \right) dr \tag{24}$$

$$= \frac{1}{\epsilon} \int_{\tau \in D_\theta} \nabla_\theta \pi_\theta(\tau) \left( R^0(\tau_r) - R^0(\tau_\epsilon(\theta)) \right) d\tau \tag{25}$$

$$= \mathbb{E}_{\pi_\theta} \left[ \nabla_\theta \log \pi_\theta(\tau)(R^0(\tau) - R^0(\tau_\epsilon(\theta))) \mid R^m(\tau) \geq Q_\epsilon^m(\theta) \right] \tag{26}$$

The second-to-last step implicitly uses the change-of-variables formula $p_\theta^m(r) = \pi_\theta(f(r)) |\det Df(r)|$, where $f \colon R \mapsto \Omega$ is the inverse function that maps rewards to $\tau$, and the fact that the determinant of Jacobian does not depend on $\theta$.

**Multi-dimensional case.** For the case where $\tau$ is an $n$-dimensional random variable, we adapt the proof of Tamar et al. (2014, Proposition 2), except for two differences: 1) the reward $R^0$ being

optimized is different from the reward $R^m$ used in the conditional expectation; 2) we condition on the outcomes within the top $\epsilon$ quantile, i.e. $R^m(\tau) \geq Q_\epsilon^m(\theta)$, rather than the outcomes below the $\epsilon$-Value-at-Risk which would be $R^m(\tau) \geq Q_\epsilon^m(\theta)$. We use the same assumptions as Tamar et al. (2014, Assumptions 4 and 5)

Define the set $D_\theta := \{\tau \colon R^m(\tau) \geq Q_\epsilon^m(\theta)\}$, which decomposes into $L_\theta$ components $D_\theta = \sum_{i=1}^{L_\theta} D_\theta^i$ (Tamar et al., 2014, Assumption 4). Let $\mathbf{v}$ denote the vector field of $\frac{\partial \tau}{\partial \theta}$ at each point of $D_\theta$. Let $\omega := \pi_\theta(\tau) R^0(\tau) d\tau$ and $\tilde{\omega} := \pi_\theta(\tau) d\tau$.

For every $\tau \in \partial D_\theta^i$, we have either (a) $R^m(\tau) = Q_\epsilon^m(\theta)$ or (b) $R^m(\tau) > Q_\epsilon^m(\theta)$. Let $\partial D_\theta^{i,a}$ and $\partial D_\theta^{i,b}$ be the subset of $\tau$ corresponding to cases (a) and (b), respectively. By the same reasoning in Tamar et al. (2014), we have

$$\int_{\partial D_\theta^{i,b}} \mathbf{v} \lrcorner \omega = 0 \,. \tag{27}$$

By definition of $D_\theta$, we have

$$\epsilon = \int_{D_\theta} \tilde{\omega} \,. \tag{28}$$

Taking the derivative, and using eq. (27), we have

$$0 = \sum_{i=1}^{L_\theta} \left( \int_{\partial D_\theta^{i,a}} \mathbf{v} \lrcorner \tilde{\omega} + \int_{D_\theta^i} \frac{\partial \tilde{\omega}}{\partial \theta} \right) \,. \tag{29}$$

In the boundary case $\tau \in \partial D_\theta^{i,a}$, $\tau$ satisfies $R^m(\tau) = Q_\epsilon^m(\theta)$, so we can denote it by $\tau_\epsilon$ as defined above. By definition of $\omega$ and linearity of the interior product, we have

$$\int_{\partial D_\theta^{i,a}} \mathbf{v} \lrcorner \omega = R^0(\tau_\epsilon)(\theta) \int_{\partial D_\theta^{i,a}} \mathbf{v} \lrcorner \tilde{\omega} \,. \tag{30}$$

Plugging eq. (29) into eq. (30), we get

$$\sum_{i=1}^{L_\theta} \int_{\partial D_\theta^{i,a}} \mathbf{v} \lrcorner \omega = -R^0(\tau_\epsilon) \sum_{i=1}^{L_\theta} \int_{D_\theta^i} \frac{\partial \tilde{\omega}}{\partial \theta} \,. \tag{31}$$

Our objective eq. (8) can be written as

$$J_\epsilon(\theta) = \mathbb{E}_\theta \left[ R^0(\tau) \mid R^m(\tau) \geq Q_\epsilon^m(\theta) \right] \tag{32}$$

$$= \int_{\tau \in \Omega} R^0(\tau) \pi_{\tau \mid R^m(\tau) \geq Q_\epsilon^m(\theta)}(\tau) d\tau \tag{33}$$

$$= \frac{1}{\epsilon} \int_{\tau \in \Omega} R^0(\tau) \pi_\theta(\tau, R^m(\tau) \geq Q_\epsilon^m(\theta)) d\tau \tag{34}$$

$$= \frac{1}{\epsilon} \int_{D_\theta} \pi_\theta(\tau) R^0(\tau) d\tau \tag{35}$$

$$= \frac{1}{\epsilon} \sum_{i=1}^{L_\theta} \int_{D_\theta^i} \pi_\theta(\tau) R^0(\tau) d\tau \,. \tag{36}$$

Its gradient is

$$\nabla_\theta J_\epsilon(\theta) = \frac{1}{\epsilon} \sum_{i=1}^{L_\theta} \nabla_\theta \int_{D_\theta^i} \pi_\theta(\tau) R^0(\tau) d\tau \,. \tag{37}$$

By the Leibniz rule, we have

$$\nabla_\theta \int_{D_\theta^i} \pi_\theta(\tau) R^0(\tau) d\tau = \int_{\partial D_\theta^i} \mathbf{v} \lrcorner \omega + \int_{D_\theta^i} \frac{\partial \omega}{\partial \theta} \tag{38}$$

$$= \int_{\partial D_\theta^{i,a}} \mathbf{v} \lrcorner \omega + \int_{D_\theta^i} \frac{\partial \omega}{\partial \theta} \,, \tag{39}$$

where the last equality follows from eq. (27). Using eq. (31) and eq. (39) in eq. (37), we get

$$\nabla_\theta J_\epsilon(\theta) = \frac{1}{\epsilon} \sum_{i=1}^{L_\theta} \left( \int_{D_\theta^i} \frac{\partial \omega}{\partial \theta} - R^0(\tau_\epsilon) \int_{D_\theta^i} \frac{\partial \tilde{\omega}}{\partial \theta} \right) \tag{40}$$

$$= \frac{1}{\epsilon} \int_{D_\theta} \nabla_\theta \pi_\theta(\tau) \left( R^0(\tau) - R^0(\tau_\epsilon) \right) d\tau \tag{41}$$

$$= \mathbb{E}_{\pi_\theta} \left[ \nabla_\theta \log \pi_\theta(\tau) \left( R^0(\tau) - R^0(\tau_\epsilon) \right) \mid R^m(\tau) \geq Q_\epsilon^m(\theta) \right] \tag{42}$$

$\square$

**Proposition 2.** *Let $R^0$ and $R^1$ be random variables related by error distribution $N$: $R^1 := R^0 + N$. Let $Q_\epsilon^0$ and $Q_\epsilon^1$ be the $(1-\epsilon)$-quantiles of the distributions of $R^0$ and $R^1$, respectively. Then*

$$P(R^0 \geq Q_\epsilon^0, R^1 \leq Q_\epsilon^1) = \epsilon \mathbb{E} \left[ F_N(Q_\epsilon^1 - R^0) \mid R^0 \geq Q_\epsilon^0 \right] \tag{10}$$

*where $F_N(r)$ is the CDF of the error distribution.*

*Proof.*

$$P(R^0 \geq Q_\epsilon^0 \wedge R^1 \leq Q_\epsilon^1) \tag{43}$$

$$= P(R^0 \geq Q_\epsilon^0 \wedge R^0 + N \leq Q_\epsilon^1) \tag{44}$$

$$= \int_{r \geq Q_\epsilon^0} P(R^0 = r, N \leq Q_\epsilon^1 - r) dr \tag{45}$$

$$= \int_{r \geq Q_\epsilon^0} P(R^0 = r) P(N \leq Q_\epsilon^1 - r) dr \tag{46}$$

$$= \int_{r \geq Q_\epsilon^0} P(R^0 = r) F_N(Q_\epsilon^1 - r) dr \tag{47}$$

$$= \epsilon \int_{r \geq Q_\epsilon^0} \frac{P(R^0 = r)}{P(R^0 \geq Q_\epsilon^0)} F_N(Q_\epsilon^1 - r) dr \tag{48}$$

$$= \epsilon \int_r \frac{P(R^0 = r, R^0 \geq Q_\epsilon^0)}{P(R^0 \geq Q_\epsilon^0)} F_N(Q_\epsilon^1 - r) dr \tag{49}$$

$$= \epsilon \mathbb{E} \left[ F_N(Q_\epsilon^1 - R^0) \mid R^0 \geq Q_\epsilon^0 \right] \tag{50}$$

$\square$

**Proposition 3.** *Let $J_{risk}$ be the Risk-Seeking Policy Gradient objective:*

$$J_{risk}(\theta) := \mathbb{E}_\theta \left[ R^0(\tau) \mid R^0(\tau) \geq Q_\epsilon^0 \right] \tag{11}$$

*and $J_{RSEP}$ be the RSEP objective (Equation 7). Given Assumption 1, optimizing for the RSEP objective, in the limit of infinite exploration, corresponds to optimizing for the risk-seeking objective.*

*Proof.* We show that for both cases of Assumption 1, we have $\tau.r = R^0(\tau)$ and $Q_\epsilon^m = Q_\epsilon^0$ in the limit.

**For Case 1**: Since all sequences have a non-zero probability of being evaluated in $R^0$ regardless of their reward values in the lowest fidelities, in the limit of infinite exploration we have that $\forall \tau, \tau.r = R^0(\tau)$ and $Q_\epsilon^m = Q_\epsilon^0$. This holds because, eventually, all samples will be evaluated in $f = 0$ regardless of their reward values due to the random sampling component, permanently replacing $\tau.r$ with $R^0$ values.

**For Case 2**: We show that RSEP will eventually only train on samples $\tau$ that satisfy $R^0(\tau) \geq Q_\epsilon^0$. First, by Assumption 1, for any fixed batch of samples, we have the inequality among the empirical quantiles:

$$Q_\epsilon^0 \leq Q_\epsilon^m. \tag{51}$$

Now we enumerate all cases that may arise during the evaluation of (9).

1. $R^0(\tau) < Q_\epsilon^0$ and $\tau.r \geq Q_\epsilon^m$. This means it mistakenly passes the multi-fidelity risk-seeking filter. However, due to passing the filter, we will have $\tau.r = R^0(\tau)$ subsequently. This means that this sample will never pass the filter on subsequent evaluations of the same batch because $R^0(\tau) < Q_\epsilon^0$ and (51).

2. $R^0(\tau) \geq Q_\epsilon^0$ and $\tau.r \geq Q_\epsilon^m$. This case is correct since $\tau$ is supposed to contribute to the gradient and it does so by passing the filter. Also note that we will have $\tau.r = R^0(\tau)$ after the gradient computation.

3. $R^0(\tau) < Q_\epsilon^0$ and $\tau.r < Q_\epsilon^m$. This case poses no issue since $\tau$ is not supposed to contribute to the gradient and it does not do so due to failing to pass the filter.

4. $R^0(\tau) \geq Q_\epsilon^0$ and $\tau.r < Q_\epsilon^m$. If this case persists across training, then $\tau$ will never be used in the gradient computations even though it should. So we need to show that this case eventually stops arising. This case occurs only if there exists another $\tau'$ that is wrongly accepted into the quantile: i.e., $R^0(\tau') < Q_\epsilon^0$ and $\tau'.r = R^{f \neq 0}(\tau') \geq Q_\epsilon^m$, which is case (1) above. However, we have shown that scenario (1) eventually does not arise, which guarantees that this scenario eventually does not arise.

Therefore, only scenario that persist are scenarios (2) and (3), which are correct. Performing a simple substitution in Equation 7:

$$\lim_{\text{training}} J_{\text{RSEP}} = \mathbb{E}_\theta \left[ R^0(\tau) \mid R^0(\tau) \geq Q_\epsilon^0 \right] = J_{\text{risk}} \tag{52}$$

Therefore, by learning using RSEP we are, in the limit, optimizing for the risk-seeking policy gradient objective. $\square$

## C   Full Empirical Results in Symbolic Regression

Table 4 and Figure 4 depict the results for all benchmarks in our baseline comparison experiment. Qualitatively, the results are the same as the ones shown in the main text. *Sequential* very clearly outperforms the other baselines by rankings as the best algorithm in 11 and 10 of the benchmarks in *max* and *avg* metrics, respectively.

Table 4: The results represent the best (max) and the average (avg) quality of samples in the hall of fame by the end of the training process. Averages across 150 repetitions. Best results for each metric are highlighted in bold.

| Benchmark | Lower bound | | Upper bound | | Shuffled | | Sequential | |
|---|---|---|---|---|---|---|---|---|
| | Max | Avg | Max | Avg | Max | Avg | Max | Avg |
| Nguyen-1 | 0.773 | 0.555 | 0.932 | 0.828 | 0.985 | 0.855 | **0.989** | **0.881** |
| Nguyen-2 | 0.844 | 0.596 | 0.901 | 0.815 | 0.927 | 0.774 | **0.945** | **0.836** |
| Nguyen-3 | 0.881 | 0.681 | 0.903 | 0.792 | 0.926 | 0.786 | **0.934** | **0.838** |
| Nguyen-4 | 0.884 | 0.703 | 0.890 | 0.788 | 0.923 | 0.786 | **0.925** | **0.846** |
| Nguyen-5 | 0.530 | 0.257 | 0.705 | 0.511 | 0.728 | 0.505 | **0.754** | **0.563** |
| Nguyen-6 | 0.800 | 0.578 | 0.918 | 0.820 | 0.966 | 0.839 | **0.969** | **0.859** |
| Nguyen-7 | 0.448 | 0.335 | 0.933 | **0.831** | 0.945 | 0.522 | **0.947** | 0.616 |
| Nguyen-8 | 0.322 | 0.275 | 0.827 | **0.660** | 0.856 | 0.519 | **0.877** | 0.608 |
| Nguyen-9 | 0.396 | 0.300 | 0.832 | 0.702 | 0.875 | 0.720 | **0.889** | **0.761** |
| Nguyen-10 | 0.498 | 0.355 | 0.851 | 0.726 | 0.872 | 0.712 | **0.883** | **0.744** |
| Nguyen-11 | 0.324 | 0.257 | 0.854 | 0.707 | **0.870** | 0.727 | 0.862 | **0.732** |
| Nguyen-12 | 0.526 | 0.366 | 0.706 | 0.561 | 0.758 | 0.505 | **0.777** | **0.621** |

Likewise, our experiment with the proposed multi-fidelity algorithms have similar results as the partial experiment shown in the main text. Table 5 and Figure 5 show that the ranking for each metric is: (i) *max*: *RSEP* with 8 wins, *RSEP_0* with 3 wins, and *PGEP_u* with 1 win; (ii) *avg*: *RSEP* with 5 wins, *RSEP_0* with 3 wins, *PGEP_u* with 2 wins, and *PGEP* with 2 wins. The baselines were never able to beat the multi-fidelity algorithms in any of the benchmarks.

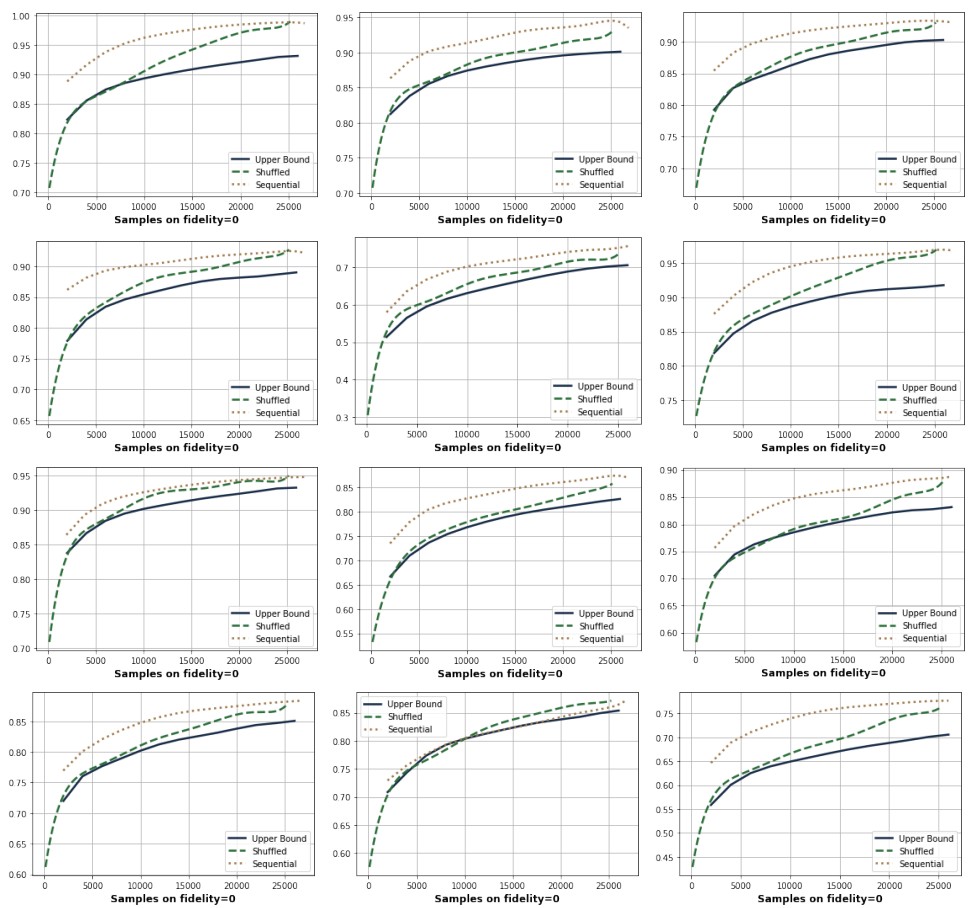

Figure 4: Average of best sample found so far during training (x-axis is the amount of samples from $\Xi^0$) across 150 repetitions. Nguyen 1-12 are depicted from left to right, top to bottom.

Table 5: The results represent the best (max) and the average (avg) quality of samples in the hall of fame by the end of the training process. Averages across 150 repetitions. Best results for each metric are highlighted in bold.

| | Sequential | | PGEP | | PGEP_u | | RSEP | | RSEP_0 | |
|---|---|---|---|---|---|---|---|---|---|---|
| Benchmark | Max | Avg | Max | Avg | Max | Avg | Max | Avg | Max | Avg |
| Nguyen-1 | 0.989 | 0.881 | 0.996 | 0.940 | 1.000 | 0.993 | **1.000** | **1.000** | 1.000 | 0.998 |
| Nguyen-2 | 0.945 | 0.836 | 0.962 | 0.915 | 0.994 | 0.955 | **1.000** | **0.980** | 1.000 | 0.957 |
| Nguyen-3 | 0.934 | 0.838 | 0.944 | 0.897 | 0.967 | 0.938 | 0.988 | 0.943 | **0.992** | **0.959** |
| Nguyen-4 | 0.925 | 0.846 | 0.946 | 0.894 | 0.956 | 0.921 | 0.985 | 0.947 | **0.991** | **0.961** |
| Nguyen-5 | 0.754 | 0.563 | 0.832 | 0.668 | 0.913 | 0.761 | **0.966** | 0.801 | 0.960 | **0.823** |
| Nguyen-6 | 0.969 | 0.859 | 0.983 | 0.944 | 0.999 | **0.981** | **1.000** | 0.965 | 0.999 | 0.942 |
| Nguyen-7 | 0.947 | 0.616 | 0.965 | **0.933** | 0.962 | 0.927 | 0.961 | 0.926 | **0.971** | 0.921 |
| Nguyen-8 | 0.877 | 0.608 | 0.903 | 0.813 | 0.937 | 0.836 | **0.976** | **0.846** | 0.912 | 0.798 |
| Nguyen-9 | 0.889 | 0.761 | 0.941 | 0.838 | 0.972 | 0.849 | **0.973** | **0.858** | 0.968 | 0.844 |
| Nguyen-10 | 0.883 | 0.744 | 0.925 | 0.858 | **0.971** | **0.901** | 0.927 | 0.862 | 0.908 | 0.819 |
| Nguyen-11 | 0.862 | 0.732 | 0.967 | **0.843** | 0.961 | 0.819 | **0.978** | 0.832 | 0.894 | 0.761 |
| Nguyen-12 | 0.777 | 0.621 | 0.786 | 0.691 | 0.796 | 0.762 | **0.792** | **0.776** | 0.795 | 0.772 |

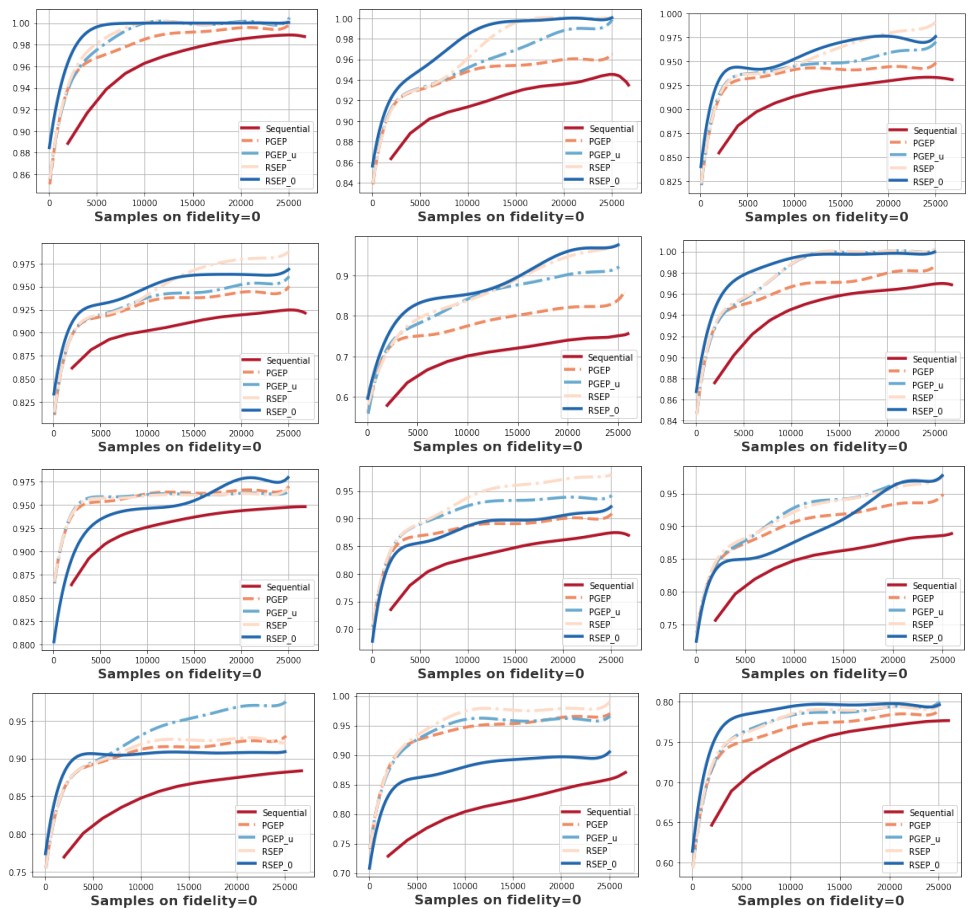

Figure 5: Average of best sample found so far during training (x-axis is the amount of samples from $\Xi^0$) across 150 repetitions. Nguyen 1-12 are depicted from left to right, top to bottom.

# D Comparison between fidelities

Figure 6 shows a comparison between our low and high fidelity simulations in the Antibody Optimization domain. While correlated, the ordering of samples are not preserved and the magnitude of the error is relatively high for this domain where picking a suboptimal antibody might result in wasting thousands of dollars in further wet lab evaluations of the candidate. This motivates the use of multi-fidelity approaches, rather than only optimizing in the low fidelity.

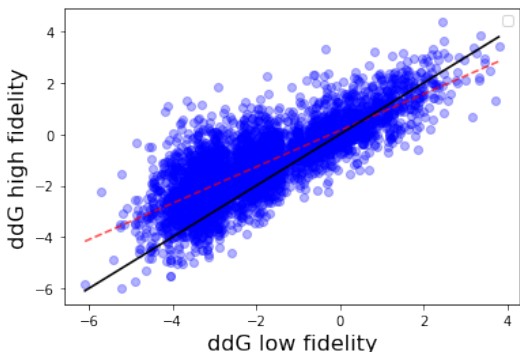

Figure 6: Comparison between fidelities in the Antibody domain. The plot contains all samples evaluated during training of all evaluated algorithms. In black we show the $x = y$ line and the dashed red line is the observed trend.

# E Full description of Symbolic Regression Benchmarks

Table 6: List of Nguyen benchmarks and their respective ground truth expressions.

| Benchmark | Expression |
|---|---|
| Nguyen-1 | $x^3 + x^2 + x$ |
| Nguyen-2 | $x^4 + x^3 + x^2 + x$ |
| Nguyen-3 | $x^5 + x^4 + x^3 + x^2 + x$ |
| Nguyen-4 | $x^6 + x^5 + x^4 + x^3 + x^2 + x$ |
| Nguyen-5 | $\sin(x^2)\cos(x) - 1$ |
| Nguyen-6 | $\sin(x) + \sin(x + x^2)$ |
| Nguyen-7 | $\log(x + 1) + \log(x^2 + 1)$ |
| Nguyen-8 | $\sqrt{x}$ |
| Nguyen-9 | $\sin(x) + \sin(y^2)$ |
| Nguyen-10 | $2\sin(x)\cos(y)$ |
| Nguyen-11 | $x^y$ |
| Nguyen-12 | $x^4 - x^3 + \frac{1}{2}y^2 - y$ |

# F Parameters for the empirical evaluation

Table 7: List of parameters used for our empirical evaluation.

| Domain | Hyperparameters |
|---|---|
| **Symbolic Regression** | General: $n_{samples} = 25,000$, $n_b = 1,000$
RSEP: $\rho = 0.1$, $\epsilon : 0.05$.
RSEP_0: $\rho = 0.05$, $\epsilon : 0.05$, $\epsilon_2 : 0.4$.
PGEP: $\rho = 0.1$, $\gamma = 0.7$.
PGEP_u: $\rho = 0.1$
Sequential: $\epsilon = 0.05$. Trains for $100,000$ samples on low fidelities.
Shuffled: prob per fidelity: $0 = 9\%$, $1 = 45.5\%$, $2 = 45.5\%$ |
| **Antibody Optimization** | General: $n_b = 72$
RSEP: $\rho = 0.01$, $\epsilon : 0.01$.
PGEP: $\rho = 0.1$, $\gamma = 0.7$.
Sequential: $\epsilon = 0.1$. Trains for $720$ samples on low fidelity. |

