# OpenReview forum: "Multi-fidelity Deep Symbolic Optimization"
_ICLR.cc/2024/Conference — ICLR 2024 Conference Withdrawn Submission_

### Official Review · Reviewer_srRU · 2023-10-31

**Soundness:** 2 fair
**Presentation:** 3 good
**Contribution:** 2 fair
**Rating:** 5
**Confidence:** 4

**Summary:**

This paper studies the problem of symbolic optimization (SO)---which aims to search over sequences of tokens to optimize a black-box reward function---in real-world applications, where the assumption that an inexpensive reward function is available does not hold. The authors introduce Multi-Fidelity Markov Decision Processes (MF-MDPs) and propose a new family of multi-fidelity SO algorithms that account for multiple fidelities and their associated costs. Experiments demonstrate the proposed method outperforms baselines on various benchmarks.

**Strengths:**

1.	The idea of introducing Multi-Fidelity Markov Decision Processes (MF-MDPs) is interesting.
2.	Experiments demonstrate the proposed method outperforms baselines on various benchmarks.

**Weaknesses:**

1.	The idea of introducing Multi-Fidelity Markov Decision Processes (MF-MDPs) is interesting. However, the proposed method to solve the MF-MDP is primarily based on some simple sampling and weighting scheme. The authors may want to explain the novelty of the proposed method in detail.
2.	The generality of the proposed method is doubtful, as the proposed method and experiments are primarily based on problems with only two reward fidelities.
3.	The authors may want to explain how to apply their method to problems with multiple (more than three) reward fidelities and transition fidelities in detail.
4.	The authors may want to evaluate their method in real-world complex problems with multiple (more than three) reward fidelities and transition fidelities.

**Questions:**

Please refer to Weaknesses for my questions.

---

> ### Author Response · Authors · 2023-11-19
>
> 1) In this paper, we primarily focused on solving a subclass of MF-MDPs that allows us to solve Symbolic Optimization problems. While our algorithms are not applicable to all classes of RL problems, there are still many applications of interest of Symbolic Optimization (for example, our main application in this paper of Antibody Optimization). Before our paper, there was no principled way of “combining” the rewards of multiple fidelities for SO problems, so one would have to either rely fully on the low-fidelity reward, or to try to refine the trained model on the high fidelity after training on the low-fidelity (our Sequential baseline). As shown in our experiments, our approach clearly beats the unprincipled way that was available before in performance, as well as providing a good theoretical understanding of when and why the algorithms work well.
>
> 2) While our evaluations and methods are more focused on the two-fidelities case (which is more common than many rewards in multi-fidelity applications), our methods are applicable to more fidelities with none or very small adaptions (for example, the symbolic regression experiment is carried out with 3 fidelities, and adding more would be possible with a slightly higher effort into setting hyper parameters correctly).
>
> 3) We cannot apply the proposed algorithms to problems with multiple transition functions (that would be a more general class of MF-MDPS that we do not focus on this paper, defining any RL problem, not only symbolic optimization problems). For applying to more fidelities, one could for example do like we did in the symbolic regression experiment (i.e., whenever “low fidelity” is read in the method, simply sample randomly from one of the low fidelities). Another domain-specific way of sampling from the low-fidelities could also be developed with ease if domain-specific knowledge is available (for example, selecting the simulation expected to be more precise for a particular token sequence).
>
> 4) Thanks for the suggestion. Developing the methods for multiple transition functions is not trivial and is a future work. We contribute in this paper a very complex real-world problem that motivated the development of those methods (antibody development), and a simple evaluation domain with 3 fidelities (Symbolic Regression).

---

### Official Review · Reviewer_LbGz · 2023-11-01

**Soundness:** 2 fair
**Presentation:** 2 fair
**Contribution:** 2 fair
**Rating:** 5
**Confidence:** 3

**Summary:**

The paper proposes to solve multi-fidelity symbolic regression using trajectory-based policy gradient algorithms. The paper introduces a few notions of MDPs specialized to their use case and propose variants of policy gradient algorithms that incorporate multiple reward functions and adapt to risk sensitive objectives. The paper shows some theoretical analysis and improvements over baseline methods.

**Strengths:**

The paper studies multi-fidelity symbolic regression using trajectory based policy gradient algorithm, which might be seen as an interesting application of policy gradient algorithm to more grounded applications. The paper also shows some promising empirical gains compared to classic baselines. This might attract and diversify RL community's attention to focus more on application rather than algorithmic research.

**Weaknesses:**

From the description of the problem setup, it feels that using sampling based optimization approach to solving multi-fidelity symbolic regression is a natural option, and hence it is not clear if the idea is very novel. Multi-objective RL and risk-sensitive RL algorithms have been extensively studied in the literature but it appears that the paper makes little comparison to those.

**Questions:**

=== **MF-MDP** ===

The MF-MDP discussed in the paper specializes to the case where all MDPs share the same transition structure, then this is effectively a single MDP with multiple reward functions. What's the benefit of introducing this extra concept of MF-MDP, since it does not seem to benefit algorithmic design in the paper. Also maybe it makes sense to borrow ideas and terminologies from the multi-objective RL literature, instead of introducing an almost identical concept.

=== **Risk sensitive policy gradient** ===

The risk sensitive policy gradient algorithm, previously studied in the RL literature such as Tamar et al, seeks to make use of the Markovian structure in the problem setup to improve the sample efficiency. This is because if we just consider the trajectory based variant of policy graident, the theoretical guarantees have become much simpler (or trivial) to derive (because the problem is effectively one-step now). Have the authors considered making use of more fine-grained structure in the sequential decision problem? From the design of the current algorithms, they are mostly trajectory based and do not leverage the MDP structure.

=== **Experiments** ===

I am not an expert in symbolic regression so cannot comment on the significance of the improvements shown in the paper. The paper does seem to compare with a fair amount of baseline approaches, but I leave to the other reviewers to comment whether such baselines are reasonable.

RSEP-0 does appear to perform the best most of the time compared to other alternatives, but not uniformly, since Fig 5 does show cases where it is under performed. Do we have a sense as to when RSEP-0 underperforms the baseline methods?

For RL experiment, it is typical that the same experiments run multiple times in order to derive the standard deviations across random initializations and optimization. This is especially the case for policy gradient algorithms where variance tends to be high. In Fig 5 no such standard deviation curve is shown.

**Details Of Ethics Concerns:**

No concerns.

---

> ### Author Response · Authors · 2023-11-19
>
> -- Weaknesses --
>
> We would like to clarify that, while somewhat related, multi objective (MO) RL and risk-sensitive (RS) RL present key differences to the Multi-fidelity case, which motivated the development of those brand-new methods. MO has multiple rewards but the algorithms try to optimize them “jointly”, which in practice makes a lot of difference than our case where we are just concerned with a single reward, but we need to use a lower-complexity reward to make sure our budget of evaluations is well-spent. RS is somewhat related in the way that it takes into account risk to guide exploration, but it doesn’t tell us how to combine multiple fidelities. We will make those distinctions more clear in the manuscript if the paper is accepted.
>
> -- Questions --
>
> 1) We apologize for not being clear in this matter, but the MF-MDP concept is used for the development of the algorithm. The main distinction from MF-MDP to any other previous type of MDP is that all the rewards estimate the same “real quantity”. Comparing to MO, for example, we are *not* interested in jointly optimizing the rewards, only the fidelity 0, and the other fidelities must be used to “guide exploration”. At the same time, it is a challenging learning problem to use the low-fidelities to train to model to guide exploration but “prioritize” the high-fidelity reward knowledge when available. While there are some intersections on relevant learning problems with other RL subareas, no other subareas have those exact same problems.
>
> 2) We thank the reviewer for the suggestion. While Risk-Sensitive PG by itself doesn’t solve the problem, we could devise a multi-fidelity algorithm inspired by it where we only explore the high-fidelity and there is low risk of getting a bad reward, or there is the potential to unexpectedly receiving a very high. We will consider the reviewer suggestion for future work,
>
> 3) We are not sure about to which experiment the reviewer refers to, as RSEP_o outperforms the sequential baseline in all of them. We would like to clarify that the other algorithms are variations of the algorithms proposed by us, and only sequential is added as a baseline to this graph because it clearly outperformed the other baselines in the previous figure.
> We removed the standard deviation from the curve because it got very hard to visualize, but we have included now to the manuscript all the error rates in a table.

---

### Official Review · Reviewer_u7RE · 2023-11-02

**Soundness:** 3 good
**Presentation:** 3 good
**Contribution:** 4 excellent
**Rating:** 8
**Confidence:** 3

**Summary:**

This paper studies Symbolic Optimization (SO) problems under the multi-fidelity setting, where the actual reward is computationally inexpensive, and lower-fidelity surrogate models or simulations can be used to speed up the learning. Instead of performing transfer learning or modifying lower-fidelity models, the authors propose to explicitly reason over the multiple fidelities. Experiments on two SO domains, Symbolic Regression and Antibody Optimization demonstrate the effectiveness of their proposed model.

**Strengths:**

1.	The authors well formulate Symbolic Optimization (SO) problems under the multi-fidelity setting, which is a quite common scenario in AI applications. The proposed novel multi-fidelity learning framework provides a good new insight.
2.	The studies on the effectiveness of different learning mode with lower-fidelities may open up a new research direction on the critical learning mode.

**Weaknesses:**

I expect the authors to report the running time of different methods, as calling the low-fidelity estimators also takes time. Moreover, it is beneficial to study the calling of different fidelities during the training time.

**Questions:**

N/A

---

> ### Author Response · Authors · 2023-11-19
>
> We apologize for being unclear regarding the running times. For the antibody optimization domain, the running time is completely dominated by the time spent evaluating samples in the high-fidelity. All the algorithms were executed for the same budget of 250 hours (this is why the graphs show a slightly different number of samples per algorithm). The time for training and sampling in the low fidelity is negligible compared to the 15 hours per batch evaluation in the high fidelity. In the symbolic Regression domain, it wouldn’t make sense to report the running time because this is a synthetic domain where we fabricate the rewards. Therefore the running times for each fidelity is artificially the same. The time taken for learning is not significantly different across the different algorithms.

---

### Official Review · Reviewer_uXaR · 2023-11-07

**Soundness:** 2 fair
**Presentation:** 2 fair
**Contribution:** 2 fair
**Rating:** 5
**Confidence:** 3

**Summary:**

This paper proposes a symbolic optimization algorithm taking into account the multi-fidelity evaluation, which is based on deep symbolic regression (DSO). In the proposed method, the promising solutions are evaluated on the highest fidelity. The experimental results on symbolic regression and antibody optimization tasks show that the proposed method can exploit the multi-fidelity information and outperform the baseline methods.

**Strengths:**

- A novel symbolic optimization method considering multi-fidelity evaluation scenarios is proposed.
- The multi-fidelity treatment in the proposed method is simple but effective on the considered tasks.
- The novelty of this paper is to introduce the multi-fidelity situation into RL-based symbolic optimization.

**Weaknesses:**

- The paper format is NeurIPS format, not ICLR style.
- The baseline methods compared with the proposed methods are very simple and limited.
- The technical novelty of the multi-fidelity treatment is unclear when comparing the existing techniques used in other algorithms.

**Questions:**

- Multi-fidelity Bayesian optimization could be a competitive algorithm. Is it possible to compare the performance of the proposed method with multi-fidelity Bayesian optimization methods? Moreover, because the proposed method is based on DSO, multi-fidelity reinforcement learning could also be the baseline.
- Could you elaborate on the technical difference and advantage of the proposed method against the existing multi-fidelity handling methods in Bayesian optimization and reinforcement learning?
- How did the authors decide and tune the hyperparameters of the proposed and baseline methods? How sensitive is the performance of each algorithm for hyperparameter settings?

---

> ### Author Response · Authors · 2023-11-19
>
> === Weaknesses ==
>
> We wish to clarify that the baseline methods added to our empirical comparison are the main general-purpose algorithms currently used as "state of the art" for symbolic optimization applications. For example, the risk-seeking policy gradient (called as upper bound or lower bound depending of the reward used) won the SRBench 2022 competition in the real-world track (https://cavalab.org/srbench/competition-2022/), with some few modifications to make it more specialized for the competition. Therefore, we are rather comparing against the state-of-the-art way of solving SO problems apart from very domain-specific solutions, instead of just comparing against very basic solutions.
>
> While those algorithms are very powerful, they assume that there is just one reward, and before our paper there was no principled way to consider the multi-fidelity scenario. "Upper bound", "Lower bound", and "Sequential" are our interpretation of the ways in which those algorithms could be applied ina. multi-fidelity problem without huge modifications, and therefore those were our baseline methods.
> == Questions ==
> 1) We haven’t implemented a Bayesian Optimization within the exact same code infrastructure (and therefore we haven’t added it to the paper because it would be hard to ensure a fair comparison). However, we also considered a BO algorithm for this antibody task in the past and it has achieved as best ddG ~= -3.5 after 1750 high-fidelity samples. Therefore, we expect that it performs worse or similarly to the “Upper bound” approach. The BO uses a sampler that weights the probability of sampling solution according to their quality in the low-fidelity, so we can consider this is a multi-fidelity BO. We are not sure about which algorithm specifically the reviewer is referring to when saying “multi-fidelity reinforcement learning”. To the best of our knowledge the “Sequential” baseline is the closest already-proposed algorithm that could be applicable to our problem (although called with different names in other publications).
>
> 2) To the best of our knowledge no RL algorithm is capable of solving MF-MDP in a principled and general manner considering that: (i) the algorithm has to take into account that the different rewards are estimations of different precisions for the same measure; (ii) there can be a mixture of fidelities in the learning batch and the algorithm has to consider that the high-fidelity takes precedence over the low fidelity. (iii) sampling in the high fidelity is very expensive and should be done strategically. Our algorithm considers all those points in a principled manner and outperforms the baselines empirically.
>
> 3) For the empirical evaluation we set the hyperparameter in a way that seemed reasonable considering the "standard Symbolic Regression" configurations in terms of batch size and epsilon value for the risk-seeking gradient set up in previous approaches. In our explorations our algorithms are not overly sensitive to hyperparameters and are only significantly affected in performance for big changes in the parameters. The parameters are fairly interpretable therefore it should be relatively easy to set them for new applications (for example, the elite-picking strategy picks a % of samples to be evaluated in the high fidelity, therefore one just has to set the % that corresponds to the number of high-fidelity evaluations they are budgeting for per batch).

---

### Official Review · Reviewer_cMbu · 2023-11-10

**Soundness:** 1 poor
**Presentation:** 2 fair
**Contribution:** 2 fair
**Rating:** 3
**Confidence:** 3

**Summary:**

The paper formulates the problem of performing Deep Symbolic Optimization (DSO) using reinforcement learning (RL) when the real black-box objective function is expensive to evaluate, but there are cheaper approximations of the objective function with different fidelities. The goal of the problem is to utilize both the real objective function and multi-fidelity approximations to find a solution that achieves a high value of the original objective function, with a limited budget for evaluations of the real objective function. With the problem formulated as an extended version of MDP with multiple versions of reward functions, the paper proposes a rule-based method to choose the reward function to use for evaluating every sampled token sequence, the main idea being "Elite-Pick", i.e., to only use the real objective function to evaluate token sequences with high potential to produce high objective value. Together with this Elite-Pick sampling rule, an RL approach RSEP, based on the Risk-Seeking Policy Gradient, and its variants are proposed to perform DSO under the problem formulation. The paper provides a brief theoretical analysis of RSEP, mostly on the approximation between the RSEP objective and the standard Risk-Seeking Policy Gradient objective in the limit of infinite exploration. The performances of the proposed RSEP and its variants are evaluated on a symbolic regression task and an antibody optimization task.

**Strengths:**

* The paper formulates an interesting problem with potential value for real-world applications: to find an efficient way of utilizing multi-fidelity approximations of the real objective function for Symbolic Optimization.

* The paper proposes a rule-based method (Elite-Pick sampling) to select the reward function to use for evaluating each token sequence sampled from the RL policy when there are different versions of the reward function with different fidelities and different associated costs. The method is based on the natural idea of only using the expensive real reward function for the token sequences with a high potential to get a high real reward value, where the potential is decided based on previous low-fidelity evaluations.

**Weaknesses:**

* The experiment results as they are now are not sufficient evidence of a performance improvement of RSEP and its variants over the baselines. The main experiment results in Figure 2 and Figure 3 show the objective function value v.s. the number of real objective function evaluations. However, the figures do not show how many low-fidelity evaluations are used for each of the compared methods. For example, **it is possible that the proposed RSEP method and its variants used more low-fidelity evaluations than the Sequential baseline in the experiments, which could also result in RSEP outperforming Sequential, but that would not be a fair comparison**. This seems quite likely to be the case for the experiments on antibody optimization, as the Sequential baseline was trained on only 720 low-fidelity evaluations before switching to only evaluating the real objective function, while Figure 3 shows that the compared methods used over 1000 evaluations of the real objective function. Using only 720 low-fidelity evaluations for Sequential when the low-fidelity evaluations are considered free while doing more evaluations on the expensive real objective function does not make much sense, and this probably indicates that RSEP consumed more evaluations than Sequential. Therefore, it would be more sensible to show the number of low-fidelity evaluations consumed by each of the compared methods as well.

* The problem formulation in the paper considers low-fidelity evaluations as free, no matter what the fidelity is. This does not support the "a whole new family of multi-fidelity SO algorithms that account for multiple fidelities and their associated costs" part of the claim in the abstract. This modeling assumption does not match the application scenarios in reality and causes a problem in the problem formulation because one could just evaluate every possible (exponentially many) token sequence using the low-fidelity approximation at the start of any algorithm. In the case where there are multiple approximations of the real objective function, it seems the zero-cost assumption would make the one approximation with the highest fidelity dominate all other approximations with lower fidelities (more on this in the first question below). As the authors point out in the future work discussion, it would be more reasonable to incorporate the costs of evaluating each fidelity into the problem formulation, where higher fidelity comes with a higher cost.

* Because the error rate is defined as $P(R^0 \ge Q^0_\epsilon, R^1 \le Q^1_\epsilon)$ instead of $P(R^1 \le Q^1_\epsilon | R^0 \ge Q^0_\epsilon)$ in Proposition 2, the first observation (Line 150) is quite trivial and does not provide much deep insight for how good the error rate is. If $\epsilon$ is reduced to a very small value, the absolute error rate will be small but we would probably care more about the conditional error rate given that a sample should pass the quantile under the real objective function. The second observation (Line 151) also only tells one side of the story, as a smaller low-fidelity quantile $Q^1_{\epsilon}$ would reduce the error rate, but at the same time, it would reduce the efficiency of using real objective function evaluations.

**Questions:**

* Given the current formulation where any approximations of the real objective can be evaluated for free regardless of their fidelity, is there any reason to evaluate a token sequence with a low-fidelity approximation when there is another approximation with a higher fidelity?

* When there are more than one non-zero fidelities such as in the experiment on symbolic regression, does the proposed Elite-Pick sampling randomly choose a non-zero fidelity regardless of their fidelities if the sampled token sequence is not over the $(1-\rho)$ quantile (Line 106 - 107)?

---

> ### Author Response · Authors · 2023-11-19
>
> -- Weaknesses --
>
> 1) We would like to clarify that 720 was selected because it represents 10 training batches, which we empirically observed that is a threshold where the algorithms are getting close to convergence but not overfitted to the low fidelity yet. If the paper is accepted we will include the results with sequential configured with many samples in low fidelity (we didn't do it yet because the experiments take over two weeks and hence there is not enough time during the rebuttal), but we don't expect the performance of the baseline to change significantly. For example, this table is the result for another target virus in which we tried the algorithm (not included in the paper), with sequential configured to 10000 samples in the low fidelity before starting to explore in the high fidelity:
>
> | Algorithm  | MAX  | Avg  |
> |------------|------|------|
> | SEQUENTIAL | 5.53 | 4.14 |
> | RSEP       | 5.61 | 4.45 |
>
> 2) We would like to clarify that the "zero-cost assumption" approximately corresponds to the reality of many applications, such as antibody optimization which inspired the development of the method. Our low-fidelity simulation takes a time in the order of ms to be completed, while the high-fidelity takes 15 hours to evaluate a single antibody sequence. Therefore the low fidelity evolution time is completely negligible compared to the high-fidelity one. While we recognize this is not the case for all applications and the rigorous development of algorithms when the zero cost assumption is not true is future work, it is still possible to use the exact same algorithms with well-thought parameters.
>
> 3) We would like to clarify that, while this Proposition 2 doesn't use extremely complex maths, it brings insights of extreme significance for the algorithm users: how to avoid the case where the algorithm gets stuck in always sampling the same samples in the high fidelity while there are better undiscovered samples that are underestimated by the low fidelity.
>
> -- Questions --
>
> 1) For non-zero fidelities, our general formulation assumes that there is no specific ordering of fidelities. For many domains (potentially the case of antibody optimization), we could choose between different simulations that give different answers but none of them is strictly better. In this setting, it might make sense to not use a single low fidelity measure, for example using domain-specific rules to sample under conditions where one simulation is expected to be better than the other. Also notice that, in cases where there is a strict ordering of simulation quality, one could adapt the algorithm to sample a fraction of best token sequences in an intermediate level simulation and a fraction of those in the high fidelity. If there is strict ordering of simulations, usually it doesn't make sense to use a low fidelity measure when a higher one is available.
>
> 2) In our experiments a random fidelity is selected. Other domain-specific strategies could be devised to select which fidelity to sample from.

---

> > ### Comment · Reviewer_cMbu · 2023-11-22
> >
> > Thanks for the authors' response.
> >
> > * My primary concern remains unaddressed: the comparison between RSEP and baseline algorithms in the experiments appears to be biased due to the differing numbers of low-fidelity evaluations used. This discrepancy challenges the paper's main conclusion that RSEP is a superior multi-fidelity Sequential Optimization (SO) algorithm. The justification for limiting low-fidelity evaluations in baseline algorithms to avoid overfitting doesn't seem adequate. A more systematic approach is needed to determine the number of low-fidelity evaluations for each method. Simply using 720 for a baseline while potentially allowing a larger number for RSEP seems arbitrary and could skew the results.
> > * The proposed method doesn't fully align with the introduction's claim of "a whole new family of multi-fidelity SO algorithms that account for multiple fidelities and their associated costs." The method assumes zero cost for all fidelity levels except the highest, which oversimplifies the scenario. It also treats different non-highest fidelity levels the same, ignoring the potential varying trade-offs among them.
> >
> > In light of these issues, my evaluation of the paper remains largely unchanged.

---

### Comment · Area_Chair_U227 · 2023-11-22

Dear reviewers,

This a reminder that deadline of author/reviewer discussion is AOE Nov 22nd (today). Please engage in the discussion and make potential adjustments to the rating and reviews.

Thank you!
AC

---

### Meta-Review · Area_Chair_U227 · 2023-12-04

**Metareview:**

This paper proposes Multi-Fidelity Markov Decision Processes and algorithms for multi-fidelity symbolic optimization. The authors have empirically showed the promising results of the proposed solution on symbolic regression and antibody optimization tasks. However I agree with the reviewers that 1) it'd be good to consider low-fidelity evaluation costs since evaluating a large number of low-fidelity data points can still be costly, 2) more evidences are needed to show the applicability of the method to more than 3 reward fidelities and transition fidelities, and as a result, 3) the technical significance is not very sufficient. These points were discussed in detail during the reviewer-AC discussion, and the majority of the reviewers still have concerns on the generality of the proposed method and the sufficiency of the experiments. Moreover, Reviewer u7RE pointed out that weakness 2,3,4 were not adequately addressed.

In the rebuttal, the authors made claims / conjectures such as "our methods are applicable to more fidelities with none or very small adaptions", "adding more would be possible with a slightly higher effort into setting hyper parameters correctly", "For applying to more fidelities, one could for example do like we did in the symbolic regression experiment", "we don't expect the performance of the baseline to change significantly". I'd suggest bringing more solid evidences before making those claims in the future to make them more convincing.

Please take a closer look at all reviews and incorporate their feedback for future submissions. Unfortunately for this conference, my recommendation is to reject this paper.

**Justification For Why Not Higher Score:**

I agree with the reviewers that 1) it'd be good to consider low-fidelity evaluation costs since evaluating a large number of low-fidelity data points can still be costly, 2) more evidences are needed to show the applicability of the method to more than 3 reward fidelities and transition fidelities, and as a result, 3) the technical significance is not very sufficient. These points were discussed in detail during the reviewer-AC discussion, and the majority of the reviewers still have concerns on the generality of the proposed method and the sufficiency of the experiments. Moreover, Reviewer u7RE pointed out that weakness 2,3,4 were not adequately addressed.

**Justification For Why Not Lower Score:**

n/a

---

### Decision · Program_Chairs · 2024-01-16

Reject